

# Acute effect of three functional fitness training designs with equalized load on inexperienced and experienced athletes

Alejandro Oliver-López[1,2], Adrián García-Valverde[3] and Rafael Sabido[1]

[1] Sport Research Center, University Miguel Hernández, Elche, Comunidad Valenciana, Spain
[2] Department of Sports Sciences, Faculty of Medicine, Health and Sports, Universidad Europea de Madrid, Madrid, Spain
[3] Faculty of Health Science, International University Isabel I of Castilla, Burgos, Castilla y León, Spain

## ABSTRACT

**Background**. In the realm of functional fitness training (FFT), three common circuits—as many repetitions or round as possible (AMRAP), for time (FT), and every minute on a minute (EMOM)—are prevalent. We aimed to elucidate the immediate impacts on athletes, considering the experience, when performing three workout modalities with matched training loads.

**Methods**. Twenty-five healthy men and women, with at least three months of experience in FFT, were allocated into the Inexperienced group (IG) and Experienced group (EG). The cut point for allocating participant in each group was set at 24 months. All of them participated in three workouts (AMRAP, FT and EMOM) with three days of rest. A double comparison was performed between level of experience (IG and EG) and among kinds of training in rating of perceived exertion (RPE), lactate concentration (LAC), countermovement jump (CMJ), heart rate (HR) and heart rate variability (HRV) using ANOVA and *post-hoc* Bonferroni tests.

**Results**. Sex was initially analyzed but had no influence, leading to combined group analyses. The workout type significantly impacted performance, with AMRAP showing differences between expertise levels ($ES = 0.81$, $p = .044$). RPE varied by workout type ($F(2,46) = 11.003$; $p < .001$), with EG reporting FT as the most and EMOM as the least demanding. Lactate levels increased across all workouts, with FT showing the highest and EMOM the lowest levels ($ES = 1.05$, $p < .001$). CMJ performance declined post-AMRAP and FT in both groups, but not after EMOM. No expertise-level differences were found in HRmean or HRmax, but HRV changes were influenced by workout type ($F(2,46) = 7.381$; $p < .01$) and expertise ($F(1,23) = 4.657$; $p = .034$), with significant decreases in HRV after AMRAP and FT for IG.

**Conclusion**. The study demonstrates that FT produced greater LAC and RPE as compared to an AMRAP, whereas EMOM generated less neuromuscular fatigue and Lac, particularly in EG. These results underscore the importance of individualizing workout selection to expertise level to optimize performance. Future research should explore longitudinal adaptation to different workout types across diverse populations.

Corresponding author
Adrián García-Valverde,
adriang.valverde@gmail.com

## INTRODUCTION

Functional fitness training (FFT) is an exercise modality that emphasizes multi-joint movements which elicit greater muscle recruitment and can be adapted to any fitness level (*Feito et al., 2018*). FFT design which combines movements from different disciplines, such as weightlifting, gymnastics and cardiovascular exercises component (*Tibana et al., 2018a*). In general, FFT sessions can be divided into three parts: warm-up, strength or skill and workout. The workout incorporates both aerobic and anaerobic training styles in intervals to improve aerobic fitness and body composition (*Dehghanzadeh Suraki et al., 2021*) and its design varies from day to day, but typically includes a mixture of high-intensity exercises of 5–20 mins. and a large number of repetitions requiring high muscular endurance and strength (*Drum et al., 2017*; *Tibana et al., 2021*). These characteristics can lead to excessive training sessions, not only for Inexperienced, but also for experienced athletes (*Weisenthal et al., 2014*).

Frequently, many FFT trainers prescribe workouts whose structure is based on a given time frame to complete all exercises (as many rounds and repetitions as possible, AMRAP) or prompt athletes to do the workout as fast as possible (for time, FT), thus both instructions often focus on the overall timing of the workout (*Mullins, 2015*). Conversely, every minute on a minute (EMOM), which is considered as interval training, is characterised by its design in time windows during which athletes are asked to keep a pace per round, although they might reach a higher recovery time if they perform faster (*Da Silva-Grigoletto, Heredia-Elvar & de Oliveira, 2020*).

However, this assumption of workout structures raises some concerns about the efficiency and effort self-regulation among athletes depending on their performance levels, since the workout intra-schedule would make it complicated to estimate optimal intensity (*Mangine & Seay, 2022*). In this sense, for the same number of repetitions, the time under tension in lower-level athletes will be greater, with the consequent repercussions on fatigue, thus conditioning the possibility to maintain the execution speed during successive repetitions (*Mazzetti et al., 2007*). Consequently, these athletes might show higher change in physiological variables like lactate concentration, heart rate and its variability as result of higher demand while fatigue might be show as decrease as jump ability (*Maté-Muñoz et al., 2017*; *Tibana et al., 2018a*; *Tibana et al., 2018b*). In connection with the self-regulation effort that each kind of workout allows, different responses and coping strategies have been noted, so rate of perceived effort (RPE) might differ depending on athlete's experience and kind of workout (*Maté-Muñoz et al., 2022*; *Oliver-Lopez, Garcia-Valverde & Sabido, 2022*).

In AMRAP, athletes perform the maximum repetitions, and they are the ones who split their breaks during training. This modality offers all athletes equal work time, but the intensity and volume of load will depend on each one of them. According to *Bellar et al. (2015)*, AMRAP is an interesting choice should the athlete be an inexperienced. This is so due to the fact that participants, regardless of their prior athletic background, exhibit a substantial physiological and cardiovascular response in terms of both aerobic fitness and anaerobic power during this workout (*Dias et al., 2022*; *Meier, Sietmann & Schmidt, 2022*).

However, in FT, athletes must perform the set of repetitions in the shortest time usually combining weightlifting and bodyweight exercises, although only bodyweight may be used in some workouts. As long as exercises were performed using only bodyweight, athletes with a higher relative strength will have a lower loss of velocity in repetitions and higher performance, effectively delaying fatigue (*Butcher et al., 2015a*). FT training will emphasise the differences between experienced and inexperienced athletes, as here everyone will perform the same number of repetitions. Responses in neuromuscular fatigue are remarkable in this type of distribution (*de Sousa-Neto et al., 2022*).

Lastly, in EMOM, inexperienced athletes will have a higher relative load at the set intensity, and they will have less rest time to start the next round of the minute, which will result in greater fatigue levels during the workout. The work-rest ratio within the minute will be the key element to differentiate training intensity, due to the fact that recovery periods are essential to avoid muscle fatigue and injury, especially in participants that are only starting FFT (*Maté-Muñoz et al., 2018*).

Despite the fact that many studies have measured acute responses to different types of workouts, only a few have addressed the specific differences that may exist among athletes' responses as a function of the workout distribution with homogenized loads (*Toledo et al., 2021*) and there are none comparing AMRAP and FT with the EMOM distribution (*Forte et al., 2022*; *Timón et al., 2019*), which is the novelty addressed in this study. Therefore, knowledge of the potential difference between kinds of workout might provide fitness coaches with a better understanding of how to optimize training programs for athletes and selecting the most appropriate workout type based on the experience level and goals.

Thus, in the present study we aimed to describe specific cardiovascular, metabolic and perceived exertion responses that experienced and inexperienced participants in FFT developed towards the three types of training, with a similar volume and relative load but organized in different workout modalities according to experience level.

## MATERIALS & METHODS

### Participants

Twenty-five people (15 males and 10 females) who practiced FFT and had at least three months of experience took part in this study. The most experienced participants had up to 36 months of experience (Table 1). We only excluded those participants who had: (i) less than three months of experience and/or less than three sessions per week in FFT ; (ii) metabolic disorders, cardiovascular diseases or musculoskeletal injuries that could affect testing or training (*e.g.*, muscle strains, contusions, *etc.*); (iii) taken drugs or supplements that would affect the measurements (*e.g.*, benzodiazepine or caffeine); (iv) not rested for at least 72 h before the sessions. In addition, we assigned participants with experience in FFT higher than 24 months to the Experienced group (EG). Otherwise, we assigned those with an experience lower than 24 months in this discipline to the Inexperienced group (IG) for comparison to previous studies (*Mangine & Seay, 2022*; *Menargues-Ramírez et al., 2022*). Prior to the enrolment, we informed all participants about the protocol, potential risks and they signed an informed written consent. We developed this study following the guidelines
**Table 1  Athletes' descriptive characteristics (mean ± standard deviation) according to strength level.**

|  | IG<br>(W) = 7 / (M) = 6 | EG<br>(W) = 4 / (M) = 11 |
| --- | --- | --- |
| Age (years) | W: 29.1 ± 6.1 | W: 36.7 ± 9.8 [*] |
|  | M: 28.6 ± 5.1 | M: 39.1 ± 6.9 [*] |
| Height (cm) | W: 164.1 ± 7.3 | W: 160.5 ± 3.3 [*] |
|  | M: 175.9 ± 12.2 | M: 174.5 ± 5.7 |
| Body weight (kg) | W: 67.6 ± 8.6 | W: 59.2 ± 5.5 [*] |
|  | M: 77.6 ± 20.1 | M: 78.6 ± 8.4 |
| Experience (months) | W: 16 ± 8.6 | W: 82.3 ± 52.2 [*] |
|  | M: 20.2 ± 4.3 | M: 66.1 ± 30.7 [*] |
| Push press 1RM (kg) | W: 46.3 ± 7.11 | W: 56.9 ± 3.27 [*] |
|  | M: 74.1 ± 8.61 | M: 83.5 ± 10.0 [*] |
| Ratio[1] | W: 0.69 ± 0.9 | W: 0.97 ± 0.2 [*] |
|  | M: 0.97 ± 0.1 | M: 1.07 ± 0.1 [*] |

**Notes.**

IG, Inexperienced group; EG, Experienced group; W, Women; M, Men.

[1] One-repetition maximal in push press divided by bodyweight.

[*] $p < 0.05$.

of the Declaration of Helsinki and conducted it in accordance to the research committee of the Miguel Hernández University (Ethics Committee ID: 230210121042).

## Study design

We designed an observational study. Participants performed three kinds of workouts and rested for 72 h between workout days (*Bishop, Jones & Woods, 2008*). Previously, we had assessed participants with a one-rep max on the push press movement which was performed according to NSCA guidelines (*Haff & Triplett, 2015*) to relativize load intensity in weightlifting movements. The barbell weight selected was 75% of 1RM push press as this is a limiting exercise in terms of kilograms lifted when compared to clean & jerk (*Buitrago & Jianping, 2018*; *Lake, Mundy & Comfort, 2014*). Before each workout, participants performed a warm-up of five minutes of jogging, three series of 10 repetitions of joint mobility (push up to bear, lunge with thoracic mobility and pull-downs using a bungee cord) and two series of progressive intensity including the movements in the workout (Table 2). Following this, they performed one of the designed workouts. In the first workout-session, participants carried out an AMRAP lasting 12 min. Individual performance (considered as repetitions) in this workout was used to set up the FT and EMOM modalities. In this sense, AMRAP volume and intensity were reproduced in FT and EMOM so that each participant performed the same individual load in every workout. We encouraged participants to perform as fast as possible the FT workout which was set with the same volume as they used in AMRAP. To optimize the EMOM structure, we organized the exercises and distributed the total individual volume equally across each minute while preserving the total volume achieved in the AMRAP. The specific distribution was as follows: (i) 1st minute for weightlifting movements only; (ii) 2nd minute for gymnastic

**Table 2** Descriptive of volume and intensity for each exercise in every workout.

| Exercise | AMRAP (12′) | | FOR TIME (max. 12′) | | EMOM | |
|---|---|---|---|---|---|---|
| | Volume | Intensity | Volume | Intensity | Volume | Intensity |
| Power clean | 5 rep | 75% 1RM | 5 rep | 75% 1RM | 5 rep | 75% 1RM |
| Push press | 3 rep | 75% 1RM | 3 rep | 75% 1RM | 3 rep | 75% 1RM |
| Pull up | 5 rep | Body weight | 5 rep | Body weight | 5 rep | Body weight |
| Push up | 8 rep | Body weight | 8 rep | Body weight | 8 rep | Body weight |
| Air squat | 10 rep | Body weight | 10 rep | Body weight | 10 rep | Body weight |
| Run | 150 m | Free | 150 m | Free | 150 m | Free |

Notes.

AMRAP, maximum number of repetitions in 12 min; FOR TIME, less time to complete the number of repetitions of the last week AMRAP with a Time Cap of 12; EMOM, as many minute rounds as repetitions in the AMRAP.

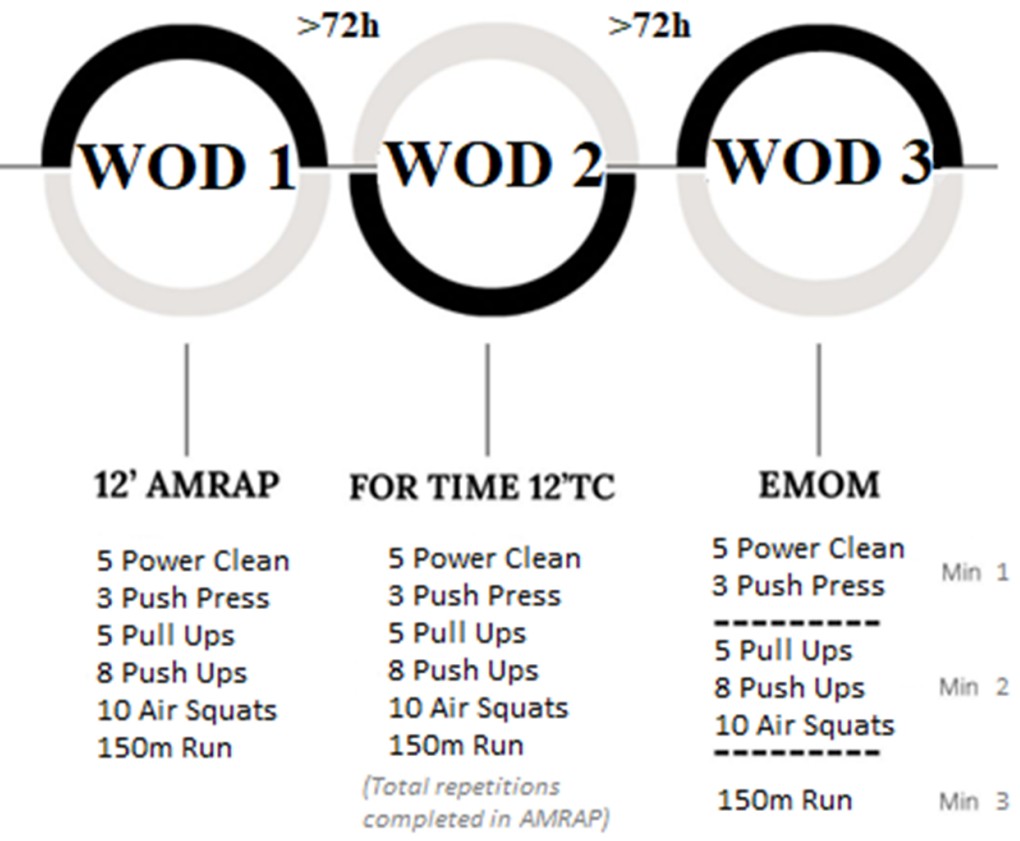

**Figure 1** Study design and timeline used to evaluate the acute effects on athletes' performance.

movements only; (iii) 3rd minute for running only (Fig. 1). All participants completed the EMOM according to distribution given.

To increase extrinsic motivation, we verbally encouraged all participants and matched them to a similarly performing opponent (*Partridge, Knapp & Massengale, 2014*). We recorded performance (repetitions in AMRAP, time to finish FT and rounds in EMOM),

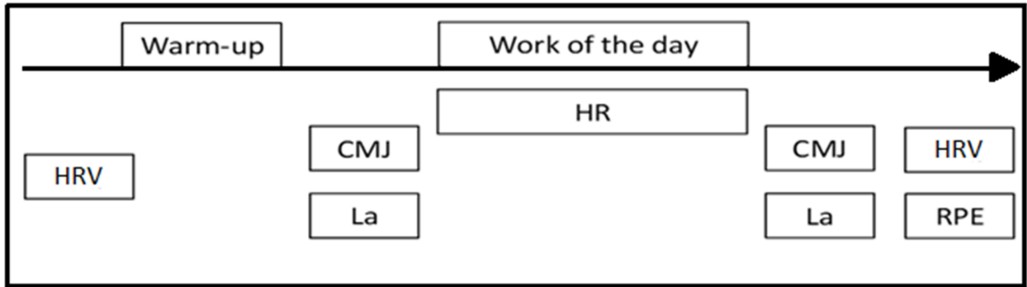

**Figure 2 Session design.** CMJ, countermovement jump; Lac, Lactate, HR, heart rate; HRV, heart rate variability; RPE, ratio of perceived effort.

rating of perceived effort, change in jump ability, blood lactate, heart rate and heart rate variability before and after each workout (Fig. 2).

## Procedures
### Performance
We performed each workout under the supervision of two qualified coaches to ensure that we met movement and workout standards. In the AMRAP, the judges recorded the athletes' total number of valid repetitions. In FT, judges decided on the number of repetitions that athletes had to perform and when completed, the judges stopped the clock to record the athletes' best time. Finally, for EMOM, judges made sure that athletes performed the number of repetitions of each minute with the correct movements and counted down (three, two, one, go) to warn participants that the next minute was starting.

### Perceived effort
We obtained the rating of perceived exertion values (RPE) by using the Borg category scale (CR-10) (*Foster et al., 2001*). We had made all athletes familiar with this scale for rating perceived exertion before the commencement of the study. CR-10 consists of a scale for exercise intensity ranging between "rest" (0) and "maximal" (10). We asked participants the following question: "How hard do you feel the workout was?", 30 min following the conclusion of the workouts to ensure that the perceived effort referred to the whole workout rather than the most recent exercise intensity (*Day et al., 2004*).

### Peripheral blood lactate concentration
We determined blood lactate concentration (LAC) by an average of test strips and a portable analyzer (Lactate Scout Plus, Biolaster, Gipuzkoa, Spain) in peripheral blood samples taken from the left earlobe, one-minute after warm-up (basal) and one minute and a half upon workout completion following the CMJ (*Tanner, Fuller & Ross, 2010*). Prior to extracting the sample, we cleaned the skin with chlorhexidine. We discarded for analysis the two-first blood drops, and the third drop was used.

### Jump ability
Jump height was assessed by Globus mat (Ergo Tester, Codognè, Italy) at the end of the warm-up and immediately after completing the workout. All participants performed six

progressive CMJs after the warm-up and were then asked to do three CMJs as high as possible with one minute of rest. They were required to keep their hands on their waist during all the test and to land with straight knees. Rebound-on landing was allowed through flexo-extension of ankles. The first contact with the mat stopped the stopwatch and the best attempt was used for analysis.

### Heart rate

Participants wore a chest band during workouts placed on the xiphoid area of the sternum with a portable device (Polar H10, Kempele, Finland), to record the heart rate (HR). We used a smartphone app (Polar Team, Kempele, Finland) to monitor several athletes at the same time. At the end of every workout, we collected the HR average (HRmean) and HR maximum (HRmax).

### Heart rate variability

Prior to the warm-up and after each workout, participants rested for 5 mins. to begin the heart rate variability assessment. Afterwards, we collected data for 1 min (*Nakamura et al., 2015*) using their chest band with a portable device (Polar H10, Kempele, Finland) connected to a smartphone with a mobile app software (Elite HRV, Asheville, North Carolina, USA), meanwhile participants were laid down on the stretcher and kept a spontaneous breath (*Saboul, Pialoux & Hautier, 2013*). We converted the indices obtained into a natural logarithm of the root mean square differences between adjacent R-R intervals (LnRMSSD) which is considered the most sensitive measure of fatigue level in a short time (*Esco, Flatt & Nakamura, 2017*). We present data as percentage of changes between pre-warm-up and post workout measurement.

## Statistical analysis

We compared performance in workouts throughout the unpaired *t*-test between two groups: Inexperienced group (IG) and Experienced group (EG) in variables of "repetitions" in AMRAP and "total time" in FT. However, we used a two-way mixed analysis of variance (ANOVA with two factors: workout and expertise level, and three and two levels, respectively) to compare the data reflecting exercise (RPE, jumping, blood lactate, heart rate and variability). We performed pairwise comparisons based on Bonferroni criterion when any main effect was observed. Besides, we checked ANOVA assumptions by normality, homoscedasticity, and independence tests (Shapiro–Wilk, Levene, and Durbin-Watson tests, respectively). We present all data as their means (M) and standard deviations (SD). In the general linear model, we calculated effect size (ES) based on Hedges' g (*Hedges, 1981*) and interpreted according to *Hopkins (2002)*. Previously to analysis, we calculated the ideal sample size to ensure enough statistical power (*Whitley & Ball, 2002*). A priori sample size analysis was based on Cohen's f large effect size (*Cohen, 1988*), setting statistical power at .8, and alpha at .05. We set statistical significance at $p < .05$. All statistical tests and plots were performed with RStudio (v4.0.2).

**Table 3  Athletes' descriptive statistics for RPE (0-10), HRmean/HRmax (bpm) and pre-post difference in CMJ (cm), HRV (RMSSD) and La+ increased (mmol/l).** All variables with mean ± SD attend to experience level. Comparisons between AMRAP *vs* FT *vs* EMOM and effect size (ES) between pre-post measures.

| | | *Inexperienced* | | | *Experienced* | | |
| --- | --- | --- | --- | --- | --- | --- | --- |
| | | **AMRAP** | **FT** | **EMOM** | **AMRAP** | **FT** | **EMOM** |
| RPE | | 7.5 ± 0.9 | 8.0 ± 0.8* | 7.0 ± 6.3* | 7.1 ± 1.1* | 7.9 ± 1.1* | 6.3 ± 0.9* |
| HRmean (bpm) | | 165.6 ± 8.2 | 164.3 ± 9.3 | 164.2 ± 6.4 | 166.1 ± 9.5 | 166.5 ± 12.7 | 162.4 ± 10.7 |
| HRmax (bpm) | | 179.2 ± 7.6 | 180.9 ± 8.5* | 178.6 ± 7.2* | 181.6 ± 12.1 | 182.7 ± 12.5 | 181.5 ± 11.9 |
| | *Pre* | 32.7 ± 7.7 | 33.2 ± 8.5 | 31.7 ± 8.3 | 37.6 ± 4.2 | 38.1 ± 4.7 | 38.6 ± 3.5 |
| CMJ (cm) | *Post* | 31.3 ± 7.6* | 31.4 ± 7.9* | 30.8 ± 7.8 | 36.1 ± 3.9* | 37.2 ± 4.5 | 37.8 ± 3.6 |
| | *ES* | (1.2) | (1.2) | (0.3) | (0.8) | (0.5) | (0.2) |
| | *Pre* | 2.3 ± 0.6 | 1.8 ± 0.6 | 1.9 ± 0.9 | 2.4 ± 0.7 | 2.4 ± 0.8 | 2.3 ± 0.8 |
| LA(mmol/L) | *Post* | 10.8 ± 2.9* | 11.9 ± 3.0* | 9.2 ± 2.8* | 10.2 ± 2.9* | 11.4 ± 4.1* | 8.1 ± 2.8* |
| | *ES* | (3.9) | (4.5) | (3.3) | (3.4) | (2.9) | (2.8) |
| | *Pre* | 61.8 ± 9.6 | 60.2 ± 7.6 | 63.5 ± 2.4 | 62.3 ± 10.7 | 57.9 ± 7.8 | 63.3 ± 9.3 |
| HRV (RMSSD) | *Post* | 48.9 ± 12* | 36.4 ± 12.1* | 36.2 ± 12.2* | 39.4 ± 16.5* | 30.7 ± 11.9* | 28.9 ± 9.6* |
| | *ES* | (1.2) | (2.3) | (2.5) | (1.6) | (2.6) | (3.5) |

**Notes.**

*$p < 0.05$.

# RESULTS

The preliminary examination of sample size revealed that a minimum of 20 participants was attained, satisfying the criterion for statistical power. Furthermore, all assumptions necessary for conducting ANOVA, including independence of observations, normality of residuals, and homogeneity of variances, were diligently verified across all variables. With these foundational prerequisites met, the subsequent analytical work aimed to delve deeper into the relationship between the kind of workout and participants' varying levels of expertise in performance, RPE, LAC, CMJ, heart rate and HRV. Firstly, we analysed the sex as a factor but no influences were found, therefore, the following analyses were performed as a single group. We report descriptive data in Table 3, sorted by kind of workout and expertise level. This description results from the idea that the kind of workout has an influence on performance depending on the participant's expertise. In this sense, the performance, considered as repetitions, in AMRAP showed differences between level groups (ES = 0.81, $p$ = .044). However, when participants performed FT based on AMRAP achievement, differences dimmed (Fig. 3).

We found no differences in RPE between experienced and beginning athletes. However, main effect in the kind of workout was showed ($F$ (2,46) = 11.003; $p$ < .001). Specifically, the expert group reported differences between workouts, pointing at FT and EMOM as the ones that required the most and the least effort (Fig. 4A), respectively. Whilst the IG only reported differences between FT and EMOM (ES = 0.97, $p$ = .02).

With respect to changes in LAC, participants showed an increment in all types of workouts (Table 3). FT registered the highest Lac, and EMOM the lowest (ES = 1.05, $p$ < .001) in both expertise groups (Fig. 4B). Additionally, these differences between workouts ($F$ (2,46) = 7.669; $p$ < .001) were more pronounced between FT and EMOM

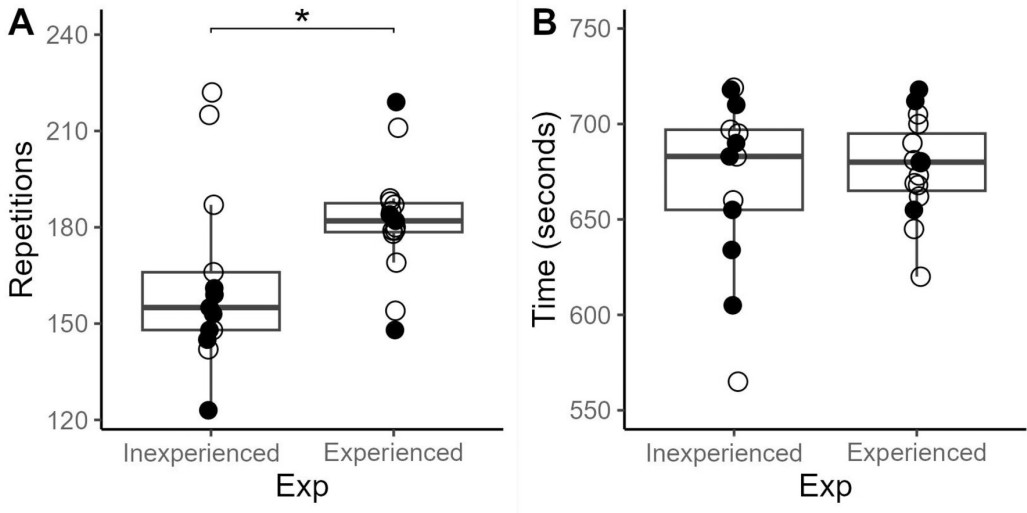

**Figure 3** **Performance in repetitions in AMRAP and finish time in FOR TIME, attending to the experience level of participants.** Empty dots mean women and fill dots men. (A) AMRAP; (B) For Time.

distributions in both expertise groups (IG: ES = 1.1, $p$ = .01; EG: ES = 1.0, $p$ = .01, respectively). However, we found no significant differences between groups when participants performed the same workout.

Participants showed worse performances in CMJ after AMRAP in both groups (IG: ES = 1.19, $p$ < .001; EG: ES = 0.81, $p$ = .013) and in IG immediately after FT workout (ES = 1.19, $p$ = .006), but this fact did not happen after EMOM. Despite the decrease, we found no differences as percentage of change between expertise groups and workout (Fig. 4C).

Regarding HR, we found no difference between expertise levels in HRmean and HRmax records through each session (Figs. 4D & 4E). However, we did not obtained differences between workout or subject's experience. Nevertheless, acute change in HRV (Fig. 4F) showed main effects in workout ($F$ (2,46) = 7.381; $p$ < .01) and expertise ($F$ (1,23) = 4.657; $p$ = .034) although *post-hoc* based on Bonferroni showed a decrease only stood out in AMRAP and FT comparison for IG (ES = 1.08, $p$ = .001).

## DISCUSSION

FFT has three typical distributions for the main high-intensity part focusing on the total of repetitions, the limit time, or the effort/rest ratio that coaches use to schedule their athletes' training (*Maté-Muñoz et al., 2018*). Currently, there are no scientific reports comparing AMRAP, EMOM and FT distributions, matching training intensity and training volume for each participant and measuring acute effects. Therefore, to our knowledge, this is the first study aimed at describing the acute responses after performing three types of trainings with a similar volume and relative intensity which were organized in different workout distributions. Concerning performance, the measurements in repetitions for AMRAP showed that the EG have a better performance despite both groups lifting an individualised load, which means that the relative load is equal for all participants. This

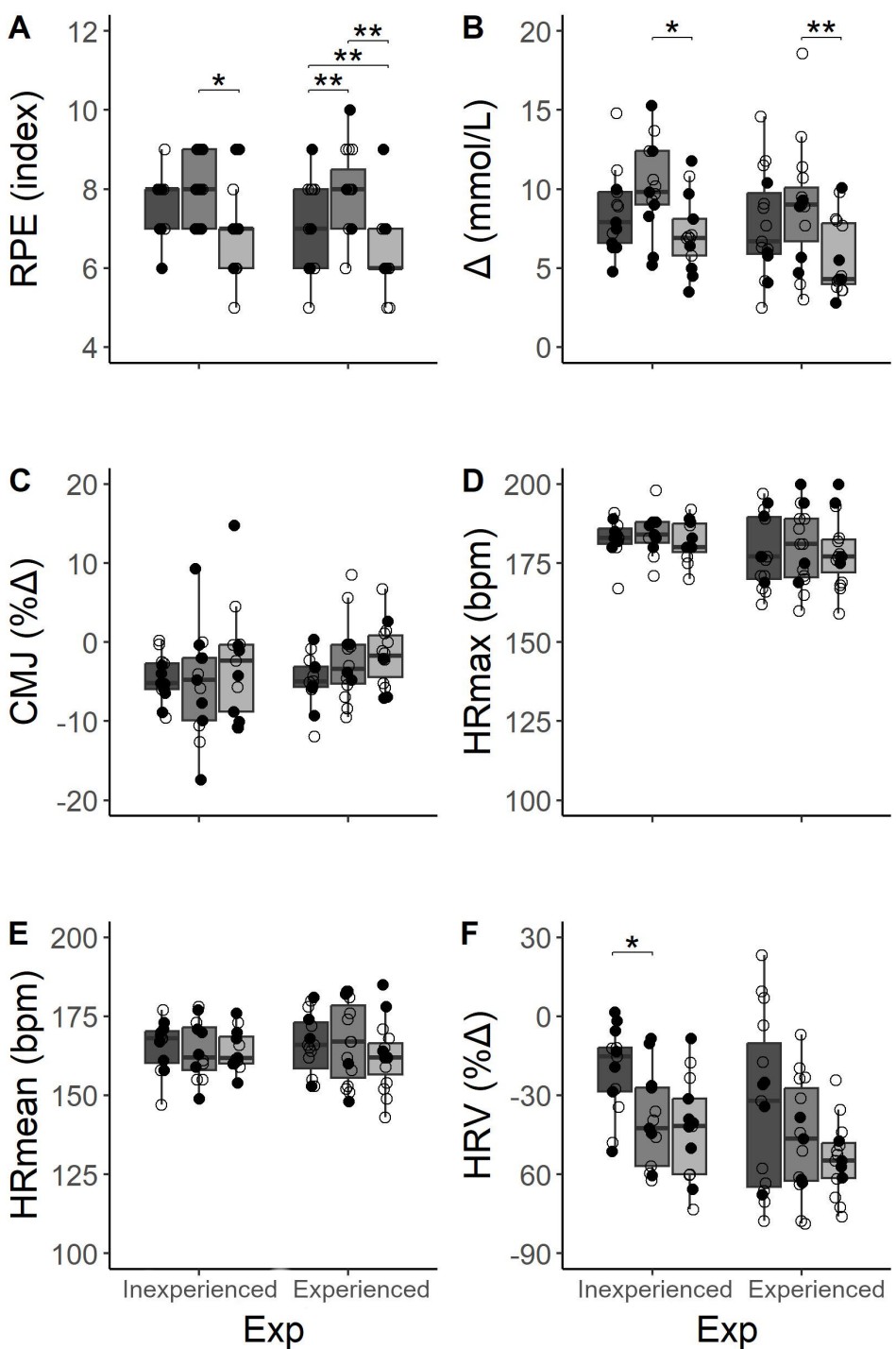

**Figure 4 Comparisons of acute measurements between three workouts distributions depending on the level of experience (Exp) in HIFT and workout type.** Dark gray, AMRAP; gray, FT; light gray, EMOM. Empty dots mean women and fill dots men.

study's results are in line with *Butcher et al. (2015a)* and *Butcher et al. (2015b)* showing that the EG performed a slightly higher number of repetitions in AMRAP, even though in this study we classified the EG as those who had only been practising for more than eight months. Moreover, our hypothesis was confirmed as the results reflected how the FT format increased the acute responses than structures such as AMRAP or EMOM. Specifically, FT elicits higher responses in lactate and RPE compared to AMRAP, while EMOM results in lower neuromuscular fatigue and lactate due to scheduled rest, especially in athletes with more than 24 months of experience in FFT.

What is more, participants reported different perceived effort depending on the type of workout although the training load was equal among them. The values manifested by the athletes are in line with the study of *Timón et al. (2019)*, in which the participants showed a perceived effort of "vigorous activity" in AMRAP, yet higher for FT training where they described its intensity as a "very hard activity". In this sense, providing a "Time Cap" to complete a standardised number of repetitions and the feedback of a coach to achieve the best time may suggest that athletes worked harder in an FT. On the contrary, athletes in the advanced group declared a perceived effort after EMOM with a mean of 6.3 over 10, which may be translated as "moderate to vigorous activity". This difference in EMOM between groups results from the EG completing the repetitions of every EMOM round in less time than the IG; they also had more time to rest until the next minute, therefore this type of work meant less effort for them (*De-Oliveira et al., 2021*). Also, an alternative explanation of these differences might be related to the knowledge of load since AMRAP could be considered as an open task (non-defined volume), whereas EMOM has structured every set.

Regarding the physiological response, all participants showed a significant difference in LAC between FT and EMOM training. Regardless of the athlete's expertise level, these results might show a scalable relationship among workouts since a similar trend could be seen in the IG and EG, with the lowest lactate levels after EMOM, followed by AMRAP and the highest lactate levels after FT. A similar study (*Timón et al., 2019*) compared AMRAP with FOR TIME and reported that experienced FFT athletes show significant differences in high lactate levels in FT with respect to AMRAP. Moreover, the fact that lactate values are in line with those in previous research, where they were reported to be around 10 mmol/L in similar workouts (*Maté-Muñoz et al., 2018*; *Tibana et al., 2016*), might support the idea that the type of workout could trigger a different physiological response. Despite this, lactate levels can be much higher depending on the muscle mass recruited by the exercises in the workout, the intensity of the participants for a training or a competition workout, or as can be seen in this work, the distribution of the training. For example, *Fernández-Fernández et al. (2015)* found up to 14 mmol/L in "Fran" and "Cindy" workouts (despite their different durations) when experienced athletes performed max effort for a personal record in a familiar workout.

Regarding neuromuscular fatigue after the three workouts, athletes reached lower height in CMJ in comparison with pre-test in AMRAP, and FT in IG, but this difference was not found in EMOM for any group. Moreover, no significant differences were identified in CMJ height loss between FFT modalities or between the experience level of the participants.

The present results contrast with *Banja et al. (2023)*, who showed a significant decrease in CMJ height after a workout, "Fran" (FT), for experienced participants. *Maté-Muñoz et al. (2018)* found the same results in this workout although this study was performed with Inexperienced participants and, in this case, the present results are in line. Moreover, these same authors saw a significant difference in the jump's height after a gymnastic workout called "Cindy" (AMRAP), but they did not report differences after an endurance workout with low-intensity cyclic movements. Eventually, in some scientific works providing evidence of responses on jump performance after a workout with weightlifting exercises, participants reduced their jump ability between an average of 3% and 5% (*Oliver-Lopez, Garcia-Valverde & Sabido, 2022*). Thus, this paper shows the same results in lower CMJ height with the exception of EMOM training, where a decrease between 1% and 3% in CMJ height would indicate that this distribution produces less mechanical fatigue due to rest intervals.

In addition, HRmax in FT reached the highest values of the three workouts in the IG with respect to EMOM although we found no statistically significant among workouts, in bpm values of HRmean and HRmax. These results agree with the scientific reports in which the load is matched for AMRAP and FOR TIME trainings (*Toledo et al., 2021*). Furthermore, the impact of these workouts on the athletes' parasympathetic nervous system resulted in a large drop in RMSSD values; the comparison among FFT modalities only showed a large difference between AMRAP and FT in IG, where the second workout modality had a greater effect on the decrease in HRV. The results in the literature are quite conclusive in terms of the acute impact on HRV of participants after workouts, but it should be noted that new research is emerging using the measurement of HRV on a day-to-day basis and how it evolves at rest (on waking up in the morning) when FFT is implemented in order to monitor an additional variable in the athletes' training load (*Castanheira et al., 2023*; *Williams et al., 2017*).

Despite the fact that there is little evidence in the literature in which several schemes of workouts are compared, we consider that the current study might raise some concerns. Firstly, we made the homogenization of workouts based on performance in AMRAP. Thus, we found an apparently light imbalance between the time of performance in AMRAP and FT (the second workout was performed faster than the first). Therefore, future studies need to compare the effect on workouts when they are homogenised based on FT or another workout and vice versa. Secondly, the EMOM structure might have had an influence on the athlete's performance since an experienced athlete's capacity allows for a higher number of repetitions. In this sense, some athletes needed to carry out additional rounds and therefore the load in each minute could have been unequal among athletes. Thirdly, we performed this research in an ecological environment, thus the study design needed adjustment to the competition calendar. This fact did not allow a more extended study, which enabled an inversion of the protocol so as to delve into the effect of creating workouts based on FT or EMOM instead of AMRAP alone.

### Practical applications

This study provides sufficient data to identify specific cardiovascular, metabolic and perceived exertion responses to three different ways of performing a workout. Therefore, coaches should be aware that training distribution is a determinant of success in the desired training stimulus for each session. The structure and feedback of the workout elicit different responses in participants and different modalities must be used depending on their experience and FFT level.

## CONCLUSIONS

This exploratory study shows that the type of workout in the three known modalities (AMRAP, FT, and EMOM) could provide the athlete with different training stimuli. Acute responses may be different depending on the type of workout for a similar load. Additionally, this study shows that FT produced a higher acute response in markers such as lactate and RPE, compared to an AMRAP regardless of athletes' level. An EMOM structure may have less neuromuscular fatigue or blood lactate due to rest time intervals, especially for experienced athletes.

## ACKNOWLEDGEMENTS

The authors acknowledge participants and technical staff for attending during the development of this study.

### Funding
The authors received no funding for this work.

### Competing Interests
The authors declare there are no competing interests.

### Author Contributions
- Alejandro Oliver-López conceived and designed the experiments, performed the experiments, analyzed the data, authored or reviewed drafts of the article, and approved the final draft.
- Adrián García-Valverde analyzed the data, prepared figures and/or tables, authored or reviewed drafts of the article, and approved the final draft.
- Rafael Sabido conceived and designed the experiments, authored or reviewed drafts of the article, and approved the final draft.

### Human Ethics
The following information was supplied relating to ethical approvals (i.e., approving body and any reference numbers):

The Miguel Hernández University granted ethical approval to carry out the study within its facilities (ID: 230210121042).

## Data Availability

The data is available at Figshare: Oliver-López, Alejandro; García-Valverde, Adrián; Sabido, Rafael (2025). DataBase_Modalities_HIFT. figshare. Dataset. https://doi.org/10.6084/m9.figshare.25858906.v1.

## Supplemental Information

Supplemental information for this article can be found online at http://dx.doi.org/10.7717/peerj.19265#supplemental-information.

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
