# Peer review of "Acute effect of three functional fitness training designs with equalized load on inexperienced and experienced athletes"

_PeerJ, doi:10.7717/peerj.19265_

## Round 0.1 · original submission · Major Revisions

The reviewers have highlighted opportunities to improve the clarity of your project's description and reporting. Please consider their feedback and make amendments as you see fit. In particular, the justification for your investigation and the methods sections have been identified as areas where there is room for significant improvement.

·

Basic reporting

Introduction
1 - Why did the authors choose the term HIFT to describe the training used, instead of other common terms like "CrossFit®" or "Functional Fitness Training"? Furthermore, the authors highlight that the "HIFT design combines movements from different disciplines, such as weightlifting, gymnastics, or exercises with a greater cardiovascular component (Tibana et al., 2018)." It is not clear in the literature whether HIFT is associated solely with metabolic conditioning or the entire training session.
See the following study:
Dominski et al., "Functional Fitness Training", CrossFit, HIMT, or HIFT: What Is the Preferable Terminology? Front Sports Act Living. 2022

2 - The authors highlighted that "However, in FT, athletes have to perform the set of repetitions in the shortest time, thus athletes with a higher relative strength will have a lower loss of velocity in repetitions and higher performance, delaying the fatigue effects (Butcher et al., 2015)"
However, not all workouts "For Time" are performed with weightlifting; therefore, this relationship between muscular strength and "For Time" needs to be reconsidered.

Methods
1 - What was the body position during the HRV analyses?
See the following study:
Holmes et al. The effects of different body positions on the accuracy of ultra-short-term heart rate variability indexes. The Journal of High Technology Management Research. 2020.

Results
1 - Why did the authors not include the absolute strength values of the power clean and push press? Even though the load performed during the workout was relative, the absolute load information allows readers to understand the caliber of the study participants.

Discussion
1 - The authors can explore the difference between AMRAP (an open task where participants do not know the total volume they can perform) and For Time and EMOM (closed tasks where participants know the total work they will perform). These differences may explain the distinct perception of effort and metabolic response between the protocols.

Experimental design

no comment

Validity of the findings

no comment

Additional comments

no comment

·

Basic reporting

The document is written in clear English and is easy to follow. Sections are well-organized, and points are coherently presented. Technical terms and abbreviations are defined and standardized, aiding comprehension even for those possibly unfamiliar with the topic. The text maintains precise and specific language suitable for a scientific context. Each claim is supported by relevant data or references, which helps eliminate ambiguities. Comparisons and study results are clearly presented. The document reflects a high degree of professionalism through appropriate technical language and structured format, adhering to academic writing conventions and the standards of the Peerj journal. It includes all expected sections of a scientific manuscript, as well as statements of ethics and informed consent. Tables and figures are relevant, high-quality, and well-described, although there is a discrepancy in terminology between Tables 1 and 3 compared to the article text, where "BG" (Beginner Group) is used in the tables, while "IG" (Inexperienced Group) is used in the text.

Experimental design

The study contributes new data and insights on the cardiovascular, metabolic, and perceived effort responses among athletes of different experience levels undergoing various types of training, which is relevant and appropriate for the scope of specialized journals in these fields. The research question is clearly defined; the study seeks to elucidate the acute impacts on athletes of different experience levels when performing three training modalities with equalized training loads. This question is relevant and significant as it addresses a gap in knowledge regarding how variations in training design can affect athletes of different levels, providing insights that may influence future training recommendations. The study adheres to the Helsinki Declaration guidelines and was approved by an ethics committee, underscoring its commitment to ethical standards. Moreover, the inclusion of informed consent and the exclusion of individuals with conditions that could affect outcomes demonstrate appropriate ethical consideration. From a technical standpoint, the use of validated tests and proper measurement equipment indicates high technical rigor. The methods are detailed, including participant selection, measurement protocols, and statistical analysis. Specific inclusion and exclusion criteria, study design details, and procedures to ensure consistency in measuring variables like perceived effort, blood lactate, and heart rate variability are mentioned. Regarding training protocol details (Lines 120 to 126), it is explained that the same total volume is maintained across the three training protocols, but the FOR TIME is designed based on AMRAP results expressed in repetitions, and the EMOM is a fixed protocol for all. How is the total volume maintained? Could this section be clarified or expanded? Also, it could detail whether all athletes completed the proposed work within each minute in the EMOM training and, if not, the procedure that followed.

Validity of the findings

The study addresses the acute response of athletes with different experiences to three common modalities of high-intensity training, which is relevant within the field of sports science and exercise physiology. However, the document does not explicitly discuss the novelty or potential impact of these findings on current practices or future research. The study is structured in a way that allows other researchers to replicate the research due to its detailed methodology and clear definition of protocols and measures. Additionally, the inclusion of details like the equalized workload setup and comparisons between groups with different experiences provides a solid foundation for future research wishing to explore similar areas or expand current findings. The document details data collection and analysis exhaustively. It describes the use of appropriate statistical analyses, including t-tests, ANOVA, and Bonferroni criteria for multiple comparisons, ensuring the statistical robustness of the results. Moreover, the data presented in tables and figures are well-organized and allow verification of claims made in the text. The study's conclusions are clearly linked to the research question, and they are confined to the data obtained from the study. The document concludes appropriately how different types of training affect athletes according to their level of experience and how these results apply to training program design. However, it could be strengthened by making a more explicit link between these findings and practical implications or future research directions in the conclusion.

Additional comments

I would like to conclude by congratulating the authors on the composition of the manuscript. The acute effects of various HIFT workouts represent a current and relevant topic that necessitates further research. Special commendation is due for the innovative proposal to include EMOM training in the comparative analysis, an approach not previously undertaken in the literature.

Summary of Issues to Resolve

1. Standardize terminology in Tables 1 and 3 and the text regarding the terms "IG" (Inexperienced Group) and "BG" (Beginner Group).
2. Explain how the volume is equalized in the three types of training protocols, if the FOR TIME training is determined based on the AMRAP training and the EMOM training is the same protocol for all athletes.
3. Detail the relationship between the obtained results and practical implications and future research lines.

Reviewer 3 ·

Basic reporting

This is a very interesting study, and I have read your paper with great interest. However, I believe there are several aspects that require significant improvements.

Abstract
The abstract should be extended and clarified, as it does not sufficiently describe the key points or findings of the study. It is unclear who participated in your study, so please provide any necessary descriptive data. When you use an abbreviation for the first time, it should be explained immediately. The results are too superficial; you should refer to the methods used and include some statistical values to support your statements. The conclusion did not consider the participants’ experience, and it should clearly state the significance of this factor.

Introduction
The introduction is generally well written, but it lacks some general information about HIFT (health, physical fitness) effects. The differences between modalities should be emphasized more thoroughly in terms of possible physiological outcomes. Then, before the last paragraph, you should indicate the gap that your study aims to address and outline the practical applications of your work.

Experimental design

Methods and Materials (M&M)
How did you assess the participants’ experience? How were they divided into groups? The study sample includes both sexes, so if men and women were assessed together, how can you ensure that this did not influence the study outcomes? The rest of the description seems clear. However, there is a concern regarding the order of the workouts, which was the same for all participants. This could mean that the effects of one training session might have influenced subsequent sessions. You should randomize the order of workouts for each participant.

Validity of the findings

Discussion
You should include a brief summary of your results at the beginning of the discussion. Since both the study aim and title refer to participants’ experience status, you need to explain this aspect in more detail.

This is an interesting study that addresses a gap in the current literature; however, the presentation of the results requires improvement.

Reviewer 4 ·

Basic reporting

The manuscript lacks clarity in articulating the study's relevance and its contribution to the field of high-intensity functional training (HIFT). The background section fails to provide sufficient context, and the literature review does not convincingly identify a gap that the study aims to fill. While the introduction briefly mentions prior studies, it does not justify why comparing the three modalities (AMRAP, FT, and EMOM) is critical. The writing, while technically accurate, would benefit from a clearer focus on the research question and a more structured argument for its importance.

Figures and tables are appropriately labeled, but the relevance of some metrics, such as CMJ (countermovement jump) and lactate, is not fully explained in the context of HIFT. The manuscript could improve by linking these variables more explicitly to the research question.

Suggested improvements:

Clearly articulate the knowledge gap being addressed and why comparing these three modalities is important.
Provide a more critical review of existing literature to highlight the study’s novelty.
Clarify the rationale for selecting the metrics (CMJ, lactate, HR, and CR10) and their relevance to HIFT performance.

Experimental design

While the protocol is detailed, the study design exhibits significant flaws that undermine the reliability of the findings. The sample size (n=25) is too small for the generalization of results, especially when divided into two groups. Although the authors conducted a power analysis, the sample lacks the diversity necessary to produce robust conclusions.

The division between experienced and inexperienced groups is problematic. The threshold of 24 months for classifying participants as experienced is not adequately justified and may fail to reflect meaningful physiological differences. The use of the CR10 scale for perceived exertion is particularly concerning, as inexperienced participants are likely to overestimate their exertion, introducing bias into the results.

Suggested improvements:

Reassess the group categorization criteria, considering more stringent thresholds for the experienced group.
Increase the sample size to improve statistical power and generalizability.
Use more objective measures of perceived exertion or supplement subjective assessments like CR10 with alternatives, such as session RPE (Foster scale), which may mitigate overestimation biases.

Validity of the findings

The findings lack sufficient validity due to methodological shortcomings. The equalization of load using AMRAP performance fails to account for individual differences in neuromuscular and metabolic responses, which could skew the comparisons between modalities. Additionally, the discussion overly generalizes results without adequately addressing key limitations, such as the small sample size and variability within groups.

The conclusions are not fully supported by the data. For instance, the claim that EMOM induces less neuromuscular fatigue for experienced athletes ignores heterogeneity within the experienced group and potential biomechanical differences. These conclusions could mislead practitioners if applied without further evidence.

Suggested improvements:

Reanalyze the data considering individual variability in performance metrics.
Address the limitations more thoroughly, including the small sample size and biases inherent to subjective measures.
Avoid generalizations in the conclusions and focus on presenting findings as exploratory rather than definitive.

Additional comments

While the study attempts to address a relevant topic within HIFT research, the execution and interpretation require substantial improvements. The focus on acute responses to different modalities is commendable, but the study does not provide actionable insights due to the issues highlighted above. Future research should consider a more robust design, including larger and more diverse samples, more objective measures, and clearer justifications for methodology and analysis.

·

Basic reporting

the present article #107157 is clear and unambiguous .it is written in professional english language .The author has done elaborative review of the literature . The framework of the article has simplified the presentation that it become easy to understand. all the relevant material is provided including raw data which made the article ethically sound . the hypotheses is missing in the article but results are well explained and discussed.

Experimental design

the primary research has a strong base and research question is well defined and tried to explore the experienced and inexperienced athletes on various bio parameters.the major flaw that is visible in the sample is of their age gap between two groups. the difference that is found between two groups, it may be due to age factor also which may be an intervening variable and may affect the result.
the difference in the age of two groups is too high .

Validity of the findings

all tools are standardized which are used in research ,so validity is not compromised.

Additional comments

the article is providing fruitful information and is beneficial for the future researchers. the only flaw is of the age diffrence between experienced and inexperienced group participants. it should be minimised to get more better clarity in the results.

---

## Round 0.2 · accepted · Accept

Thank you for your efforts in addressing the reviewer feedback. I am please to advise that you have satisfactorily addressed all feedback and your article is suitable for acceptance.

·

Basic reporting

The authors have made the necessary modifications, and it is accepted in its current form.

Experimental design

The authors have made the necessary modifications, and it is accepted in its current form.

Validity of the findings

The authors have made the necessary modifications, and it is accepted in its current form.

Additional comments

The authors have made the necessary modifications, and it is accepted in its current form.

Reviewer 3 ·

Basic reporting

All issues were addressed

Experimental design

All issues were addressed

Validity of the findings

All issues were addressed

Additional comments

All issues were addressed